# Brewing By-Product Upcycling Potential: Nutritionally Valuable Compounds and Antioxidant Activity Evaluation

**DOI:** 10.3390/antiox10020165

**Published:** 2021-01-22

**Authors:** Elisabetta Bravi, Giovanni De Francesco, Valeria Sileoni, Giuseppe Perretti, Fernanda Galgano, Ombretta Marconi

**Affiliations:** 1Italian Brewing Research Center CERB, University of Perugia, Via S. Costanzo, n.c.n., 06126 Perugia, Italy; giovanni.defrancesco@unipg.it (G.D.F.); ombretta.marconi@unipg.it (O.M.); 2Department of Economy, Universitas Mercatorum, Piazza Mattei 10, 00186 Roma, Italy; valeria.sileoni@unimercatorum.it; 3Department of Agricultural, Food and Environmental Sciences, University of Perugia, Via S. Costanzo, n.c.n., 06126 Perugia, Italy; 4School of Agricultural, Forestry, Food and Environmental Sciences, University of Basilicata, Via dell’Ateneo Lucano, 10, 85100 Potenza, Italy; fernanda.galgano@unibas.it

**Keywords:** spent grains, spent hop, phenolics, antioxidant activity, β-glucan, arabinoxylan, fiber, proteins

## Abstract

The brewing industry produces high quantities of solid and liquid waste, causing disposal issues. Brewing spent grains (BSGs) and brewing spent hop (BSH) are important by-products of the brewing industry and possess a high-value chemical composition. In this study, BSG and BSH, obtained from the production process of two different types of ale beer (Imperial red and Belgian strong beer) were characterized in terms of valuable components, including proteins, carbohydrates, fat, dietary fiber, β-glucans, arabinoxylans, polyphenols, and phenolic acids, and antioxidant activity (Ferric Reducing Antioxidant Power Assay (FRAP), 2,2-Diphenyl-1-picrylhydrazyl (DPPH) and 2,2′-Azino-bis (3-ethylbenzothiazoline-6-sulfonic acid) diammonium salt (ABTS)). Significant concentrations of total polyphenols were observed in both BSH and BSG samples (average of about 10 mg GAE/g of dry mass); however, about 1.5-fold higher levels were detected in by-products of Belgian strong ale beer compared with Imperial red. Free and bound phenolic acids were quantified using a validated chromatographic method. A much higher level of total phenolic acids (TPA) (about 16-fold higher) was found in BSG samples compared with BSHs. Finally, their antioxidant potential was verified. By-products of Belgian strong ale beer, both BSG and BSH, showed significantly higher antioxidative capacity (about 1.5-fold lower inhibitory concentration (IC50) values) compared with spent grains and hop from the brewing of Imperial red ale. In summary, BSG and BSH may be considered rich sources of protein, carbohydrates, fiber, and antioxidant compounds (polyphenols), and have the potential to be upcycled by transformation into value-added products.

## 1. Introduction

The issue of food waste disposal concerns the life cycle of food, comprising the agricultural system, industrial manufacturing and processing, as well as retail and consumption. Nowadays, companies are interested in the treatment of waste in order to upcycle it by transformation into higher-value products, with the aim of a “zero waste society” in which waste is used as raw material for new products and applications. The aim of waste management in the agri-food industry is to improve resource efficiency while protecting the environment. For this purpose, management strategies are essential for the proper reduction in waste or to transform waste into new raw materials. This kind of approach is included in the circular economy system, which is an industrial “regenerative or restorative” system aimed at eliminating waste and pollution [1,2]. Large amounts of waste are generated throughout the food chain during harvesting, transport, processing, storage, distribution and retailing. Moreover, it was estimated that around 12.9% of food waste is generated from cereal processing and manufacturing [1]. Cereal waste is a possible source of proteins, polysaccharides, antioxidants, minerals, lipids and vitamins. [1]. The main purpose of modern beer brewing is the achievement of a final product with a high quality and a wide flavor diversification [3,4]. In any case, the brewing industry produces huge amounts of solid and liquid waste that cause serious ecological and economical management concerns. The production of waste occurs at different stages of the brewing process (mashing, boiling, fermenting, filtering) and the waste processing technologies used depend on the volume, composition, and potential value of these residues [5]. The brewing industry has experienced several technological advances that allow for large savings by lowering the amount of waste and by-products produced throughout the entire brewing process. Nevertheless, certain proper by-products of beverage production, such as brewing spent grains (BSGs), residual brewing yeast and brewing spent hop (BSH) or trub, have hardly had their volumes reduced [6].

About 85% of the brewing waste produced is BSG (this amount corresponds to about 14 to 20 kg of BSG for each hectoliter of beer produced) and is composed of grain residue fractions—mainly husk, pericarp and fragments of endosperm [7]. BSG can be used for animal nutrition as a healthy feed that also allows for a reduction in feeding costs. They can also be incorporated into the human diet as a fiber supplement, such as in bread or biscuits. Moreover, BSGs are used for biogas production [8]. Sometimes, BSGs are used in the construction sector in order to reduce brick weight and to increase thermal insulation ability. BSGs can also be used for metal adsorption in wastewater, yeast immobilization, and as enzymes for ethanol, xylitol or lactic acid production. Finally, valuable and biologically active constituents, such as antioxidant compounds, can be extracted from BSG [9,10,11]. In Europe, BSGs, accounting for nearly 34–35 million tons of spent grains every year, are mainly sold to farmers as animal feed, while also being used as a feedstuff in aquaculture, and bakery product preparation [12,13,14]. The European legislation establishes limits for the marketing and distribution of BSG: up to 0.3% of dimethylpolysiloxane, up to 1.5% of enzymes and up to 1.8% bentonite [15]. BSH is the brewing term used for the material, after the wort has been boiled and separated in a whirlpool. BSH contains polyphenols, isomerized acids adsorbed on the solid trub residues and the insoluble materials of hops, as well as the condensation products of wort proteins [16].

In relation to BSH, nearly 85% of used hops will create beer by-products [5]. In general, 0.2 to 0.4 kg of BSH waste is created for every hectoliter of beer produced, and its moisture content is between 80% and 90% [3]. The amount produced is not high enough to cause management issues in relation to BSH’s disposal, but it is an interesting by-product to upcycle because of its peculiar chemical composition. It contains lipids, amino acids and coagulated proteins, and it could be used to produce high-value products. The main methods of disposal of BSH are to reuse them as fertilizer or compost, due to their high nitrogen content, or mixed with BSG as animal feed. Moreover, the essential oils of BSH have been tested for use as insect repellents [5,17]. Moreover, the valorization of brewing by-products (both BSG and BSH) could be achieved through the extraction of high-value compounds such as proteins, fibers, polysaccharides, flavor compounds and phytochemicals, which can be added as functional ingredients in the preparation of beverages and food, dietary supplements and cosmetic products [18].

The present study aimed to characterize two solid by-products of the brewing process (BSG and BSH) in terms of their components and nutritional claims, such as their antioxidant compounds (total polyphenols, and phenolic acids) and relative antioxidant activity. The objective was to contribute to the enrichment of the current knowledge and to endorse the nutritional value of brewing by-products and the potential advantages of their upcycling by formulating innovative functional foods. The content of β-glucans and arabinoxylan, fats, carbohydrates, sugars and total dietary fiber (TDF) was also determined.

## 2. Materials and Methods

### 2.1. Chemicals

Methanol, water and acetonitrile (HPLC grade), 2,2-Diphenyl-1-picrylhydrazyl (DPPH), 2,2′-Azino-bis(3-ethylbenzothiazoline-6-sulfonic acid) diammonium salt (ABTS), 2,4,6-Tris(2-pyridyl)-s-triazine (TPTZ), gallic, α-resorcylic, tyrosol, gentisic, *p*-hydroybenzoic, 2,6-dihydroxybenzoic, *m*-hydroxybenzoic, vanillic, salicylic, syringic, homovanillic, *p*-coumaric, *m*-coumaric, *o*-coumaric, ferulic, sinapic, caffeic, and chlorogenic acid were purchased from Sigma-Aldrich (St. Louis, MO, USA). All other chemicals used were of analytical grade.

### 2.2. Samples

Four solid residues of the brewing process were collected and studied: two BSGs and two BSHs, provided by a commercial brewery located in Perugia (Italy). By-products were collected immediately after their production in the respective brewing process stage and resulted from the production of an Imperial red ale (IRA) and a Belgian strong ale (BSA) beer. The IRA beer (16.5 °P) was prepared using barley malt, including Pilsner, Munich and Melanoidin malt, as well as two different hop products—CO_2_ extract (0.3 g/L) of Herkules high-alpha acid variety and granules (granules, 0.8 g/L) of the German Tettnanger (bittering and aroma variety). The BSA beer (17.5 °P) was prepared using Pilsner, pale malt, and flaked wheat, as well as three different hop products—CO_2_ extract (0.25 g/L) of Herkules variety and granules of Hallertau Magnum (0.3 g/L, bittering variety) and East Kent Golding (1 g/L, aroma variety). The BSG samples were collected immediately after lautering and the BSH after wort boiling. Both types of brewing by-products were collected from a blend of three different batches, and then carefully mixed before sampling to avoid inhomogeneity. In total, four different samples were studied—two obtained from the brewing of Imperial red ale (BSG-IRA and BSH-IRA) and two from Belgian strong ale (BSG-BSA and BSH-BSA).

### 2.3. Sample Processing

In order to stabilize the brewing by-products and to facilitate their management and storage during the entire study, it was necessary to reduce the water content of both the considered by-products. BSG samples were drained and dried in a fluidized bed dryer at the temperature of 50 °C for 1.5 h, and finally stored, in sealed polyethylene bags under vacuum, at room temperature. BSH samples were cooled, frozen, freeze-dried, and stored in dark brown glass vessels at room temperature.

### 2.4. Analytical Determination

The parameters analyzed were: moisture (relative humidity, RH%) and water activity (aw), ash, TDF, total nitrogen and proteins, fat, β-glucans and arabinoxylans, total polyphenols (TPs, sum of polyphenols in free and bound fractions), phenolic acid (PA) content, and antioxidant activity (by DPPH, FRAP and ABTS assays).

All the samples were analyzed in triplicate, except the phenolic acid composition that was analyzed in sextuplicate.

#### 2.4.1. Moisture, Nitrogen, Ash and Fat Content

Moisture, nitrogen, ash and fat content were assessed as described using AOAC methods 14.004, 945.18-B, 14.006 and AOCS 1984, Aa 4–38, respectively. Protein content was calculated using the nitrogen conversion factor (6.25) [19].

#### 2.4.2. Total Dietary Fiber

TDF was determined via a Megazyme assay kit (Megazyme International, Wicklow, Ireland) following the AOAC method 985.29.

#### 2.4.3. Carbohydrates

Carbohydrates were calculated as the percentage difference with moisture, proteins, ashes, fats and TDF.

#### 2.4.4. Water Activity

The a_w_ was determined using an Aqualab^®^ series 3 (Decagon, Pullman, WA, USA) calibrated with potassium chloride solution (a_w_ = 0.250 ± 0.003).

#### 2.4.5. β-glucans and Arabinoxylans Content

β-glucans were determined for BSG and BSH using a Megazyme assay kit (Megazyme International, Wicklow, Ireland), following the EBC method 4.16.1. Arabinoxylan content was determined as reported by Marconi et al. [20]. Briefly, 10 mL of acetate buffer was added to 500 mg of a milled sample for the solubilization of arabinoxylans. The enzymatic degradation of the interference molecules was performed by adding α-amylase (Megazyme 3000 U/mL, with stirring at 80 °C), which was followed by the addition of amyloglucosidase (Megazyme 3260 U/mL with stirring at 60 °C). Finally, pancreatin and lichenase were added (with stirring at 40 °C). Precipitation of the purified pellet was performed by the addition of pure ethanol; the supernatant fraction was discarded, and the pellet was washed with ethanol (85% *v*/*v*) and acetone. Acid hydrolysis of the pellet was carried out by adding 12 M sulfuric acid. Quantification of arabinoxylans was performed through spectrophotometric measurements, using a pentosans-specific reagent (phloroglucinol), the absorbance versus blank reagent was read at 510 nm and subtracted from the absorbance at 552 nm.

#### 2.4.6. Determination of Sugar Profiles

Free sugar extraction was carried out by homogenizing 10 g of the sample in 100 mL 80% ethanol solution with an ultra-turrax while heating in a boiling water bath for 5 min, until finally cooling the sample to room temperature and subjecting it to centrifugation at 5000 *g* for 10 min. The supernatant was vacuum evaporated and dissolved to a known volume in distilled water. The sugar profile was obtained using an HPLC-Evaporative Light Scattering Detector (ELSD) (Alltech Associates Inc., Deerfield, IL, USA) system following the method proposed by Floridi et al. [21].

#### 2.4.7. Extraction of Free and Bound-Forms of Phenolic Compounds

The free fraction was obtained following the method of Stagnari et al. [22]. Five ml of a solution of methanol, water, acetic acid (70/29.5/0.5) was added to dried BSG or freeze-dried trub (1 g); the mixture was homogenized with ultra-turrax for 1 min on ice, ultrasonicated for 40 min at room temperature and centrifuged for 10 min at 5000× *g*. The entire extraction procedure was conducted whilst avoiding exposure to light and to high temperature. The extraction was repeated twice, and the supernatants were collected, evaporated to dryness under a vacuum and dissolved in 1 mL of methanol before the determination of the TP content, and antioxidant activities (DPPH, FRAP, and ABTS). For HPLC analysis, the extract was redissolved in 1 mL of 30% methanol in eluent A (0.1 M citric acid and 0.2 M sodium hydrogen phosphate; 85:15; *v*/*v*), before the determination of the phenolic acid fraction. If necessary, to correctly quantify the amount of PAs, different dilution of extracts in eluent A were prepared. The bound forms of phenolic acids were determined after liberation by alkaline hydrolysis of the solid residues left after extraction of free phenolic forms, as reported by Ceccaroni et al. [23]. Briefly, the hydrolyses were performed with 10 mL 4 M NaOH; the suspension was left overnight at room temperature after an initial treatment of 40 min in a sonication bath. After alkaline hydrolysis, the mixture was adjusted to a pH of 2 with 6 M HCl and extracted three times with 20 mL of ethyl acetate. The three supernatants were collected, evaporated under a vacuum with a rotary evaporator, dissolved in (i) methanol for the determination of total polyphenol content and antioxidant activities (DPPH, FRAP, and ABTS), and (ii) 30% methanol in eluent A (0.1 M citric acid and 0.2 M sodium hydrogen phosphate, 85/15 *v*/*v*) before the determination of the phenolic acid fraction.

#### 2.4.8. Determination of Total Polyphenol Content

The Folin–Ciocalteau method [24] was applied to determine the TP content of the free (FP) and bound (BP) fractions extracted from BSG and BSH. In total, 2 mL of Folin–Ciocalteau reagent was added to 0.4 mL extracts, then 1.6 mL of a 7.5% Na_2_CO_3_ solution was added. The obtained solution was incubated at room temperature, in the dark, for 120 min. The absorbance of the mixture was measured at 760 nm. Standard solutions of gallic acid (GA) were used to calibrate the method, and the content of FP and BP was expressed as mg of GA equivalent (GAE) per g of sample dry matter (mg GAE g^−1^ dm).

#### 2.4.9. Antioxidant Activity

Antioxidant activity describes the ability of some compounds to reduce oxidant compounds and to scavenge free radicals. In this study, ABTS, DPPH and FRAP tests were used to measure the antioxidant ability of brewing by-products (BSG and BSH), using Trolox as a standard for the calibration curves and following the same procedures used by Benincasa et al. [25].

For ABTS, an aliquot of phenolic extract was mixed with the ABTS• ^+^ solution, and the absorbance was read at 734 nm after 2 h in the dark. For the DPPH assay, an aliquot of phenolic extract was mixed with DPPH solution, and the absorbance was read at 515 nm after 30 min in the dark. For the FRAP assay, an aliquot of phenolic extract was mixed with the FRAP working solution and warmed at 37 °C, in the dark, for 30 min. The FRAP reaction mixture of the samples was read at 593 nm. The standard curve was linear between 25 and 800 µM Trolox and the results of all tests were expressed as Trolox equivalents (TE) g^−1^ (dry basis) of the sample. The median inhibitory concentration (IC50), that is the concentration of an antioxidant-containing substance required to scavenge 50% of the initial redox/radical systems used for antioxidant evaluation, was also calculated for each antioxidant assay. The lower the IC50 value, the more potent is the substance at scavenging and this implies a higher antioxidant activity [26].

#### 2.4.10. Determination of Phenolic Acids

PAs were analyzed using an UHPLC system consisting in a Knauer 3950 autosampler with a 10 μL loop, a quaternary Azura P 6.1 L pump (Knauer, Berlin, Germany) coupled with an Azura MWD 2.1 L height channel UV–vis detector. The separation was carried out using a SunShell C18 column (ChromaNik Technologies Inc., Osaka 552-0001 Japan, 50 mm × 2.1 mm ID) at 25 °C and a flow rate of 0.4 mL/min. Mobile phase A was 0.1 M citric acid and 0.2 M sodium hydrogen phosphate (85/15; *v*/*v*), and mobile phase B was phase A, methanol and acetonitrile (30/20/50, *v*/*v*/*v*). The pH of mobile phase A was 2.88 and the pH of mobile phase B was adjusted to 3.44 with 85% orthophosphoric acid. The chromatographic separation was achieved using the following elution gradient: mobile phase A 90% (0 min), 100% (2 min), 70% (8 min), 50% (10 min), 20% (12 min), 90% (12.5 min). The wavelengths of the three channels used for the detection were 254, 278, and 324 nm, and the Clarity Chromatography Software for Windows (DataApex, Prague, Czech Republic) was used for data acquisition and elaboration. The external standard method was used for the calibration (a stock solution of 100 ug/mL of a mix of considered phenolic acids in methanol/eluent A (30:70, *v*:*v*) was used to prepare working solutions), and the calibration plots were constructed for standard compounds with a linearity between 0.5 and 5 μg/mL. The calibration curve for each phenolic compound was plotted using UV detection close to the maximum UV absorption for a given substance (270 nm for 3,4,5-trihydroxybenzoic, tyrosol, homovanillic, syringic, *m*-coumaric, and *o*-coumaric; 254 nm for 3,5-dihydroxybenzoic, 4-hydroxybenzoic, 2,6-dihydroxybenzoic, 3-hydroxybenzoic, and vanillic; 324 nm for 2,5- dihydroxybenzoic, caffeic, chlorogenic, *p*-coumaric, salicylic, ferulic and sinapic).

### 2.5. HPLC Method Validation

The validation of the method of quantitative determination of free phenolic acids in BSG samples using HPLC-UV analysis was run.

The process of validation of an analytical method is carried out to evaluate the method’s performance, to confirm that the employed method meets the intended requirements and to assure the reliability of the obtained results [27]. The validation parameters calculated were: 1. system suitability; 2. linearity; 3. homoscedasticity; 4. limit of detection (LOD) and limit of quantitation (LOQ); 5. accuracy (trueness and precision), and 6. uncertainty.

The system suitability was calculated by replicated injections (*n* = 10) of the same standard solution containing PAs and tyrosol [27]. The parameters used in the system suitability test were the relative standard deviations (%RSD) of retention time and peak area for each PA, the number of theoretical plates (N), and the tailing factor (T).Linearity of the method was tested in the PA concentration range of 0.5–5 µg mL^−1^. The solutions were chromatographed ten times, in accordance with the Eurachem Guide [28]. External calibration curves were constructed for each PA at six calibration levels, and the correlation coefficient for each plot was calculated.The homoscedasticity of the calibration plots was verified using Cochran’s test. In Cochran’s test, the test statistic calculated is the ratio of the largest variance to the sum of all variances (Cc, C calculated) [29].The limit of detection (LOD), determined as three times the signal-to-noise ratio, and limit of quantitation (LOQ), determined as ten times the signal-to-noise ratio, of each PA were calculated and are reported in Table 1 [30].The trueness (how close the mean of ten replications is to a reference value) was calculated, together with the precision (how close results are to one another) in order to investigate the accuracy of the results obtained with the developed method. The trueness was calculated to compare the difference between mean spiked BSG samples and the mean value with the added concentration of each analyte, and was expressed as the relative spike recovery, R’% [31]. The precision was evaluated by analyzing ten times the BSG fortified samples. Precision is expressed by statistical parameters (such as relative standard deviation, RSD) that describe the spread of results and is a measure of how close results are to one another. The relative standard deviations of repeatability (%RSDr) were calculated.The uncertainty of the developed method was tested by calculating the Horwitz ratio value (HorRat). The HorRat value is a variation of the Horwitz equation to be used in a single-laboratory validation study [30].

### 2.6. Statistical Analysis

The statistical analyses were performed with SigmaPlot software (SPSS Inc., Chicago, IL, USA) version 12.5. Significant variations between the different samples were discriminated using Student’s *t*-test analysis (*p* ≤ 0.01).

## 3. Results and Discussion

### 3.1. Brewing By-Product Characterization

The scientific research has shown that the chemical composition of BSG always includes considerable amounts of valuable nutritional compounds, such as dietary fiber, protein, minerals, polyphenols, vitamins and lipids [18,31]. The average of the results and the respective standard deviations (based on dry matter) for the composition of the total fraction of brewing by-products are reported in Table 2.

The moisture content of fresh products was about 80% for all BSG and BSH samples, and the ash values on dry bases ranged between 2.11 and 2.96, which were consistent with the literature data [7,11,31]. The a_w_ values of dried samples were 0.35 and 0.42, indicating the stability and safety of the dried by-products with respect to microbial growth, rates of deteriorative reactions and chemical/physical properties [32]. In general, the chemical composition of brewing by-products (spent grains and spent hop) depends mainly on raw ingredients used, such as type of grains and malt and hop products, as well as the brewing process itself (type of beer produced). The concentration of proteins in spent grain depends mainly on the type and quality of grain (barley, wheat or other), harvest time, milling, malting and mashing conditions; in addition, the clarification step and the type and quality of other secondary raw materials are important. The amount of protein in BSH depends mainly on the brewing process, the type of malt or other grain and the hop product used (some hop products, such as CO_2_ extract, may have a lower or no concentration of proteins; others, such as pellets, may have a higher content) [10,17,33]. In our study, the amount of protein in BSG was relevant, at 19% and 16% (dry basis) for IRA and BSA, respectively. The protein content for BSH was higher, at 52% and 40%. These data are consistent with the intrinsic characteristics of trub, which is known as denatured protein clot precipitate after wort boiling [34]. The obtained results fitted with the data in the literature [7,31]. The protein content in the residual BSG was calculated using the conversion factor of total nitrogen into crude protein (6.25). The same factor was used for the BSH, considering that the main sources of protein in trub are the brewed grains (in our experimentation, these were malt and wheat) [35].

Owing to their protein-rich composition, BSG and BSH have the potential to be employed to improve human nutrition, as in the case of whey proteins [36,37].

The fat content of BSG, as reported in Table 2, was 6.8% and 6.4% for IRA and BSA samples, respectively. The content of fat in BSH was lower, accounting for around 1% for both the trub samples, in agreement with the data in the literature [7,18,31,38]. Carbohydrates in BSG and BSH, calculated as the percentage difference between different components of the sample, were present in relevant concentrations for brewing by-products. Del Rio et al. [39] found similar concentrations of carbohydrates in BSG. Forssell et al. [37] reported an even higher amount (45%). In any case, the content of carbohydrates during different steps of the brewing process depends on the barley or other cereal varieties, as well as malting, and greatly relies on the brewing process, particularly on the lautering step. The amount of carbohydrates present during different steps of the brewing process depends on the barley or other cereal varieties, as well as the malting and brewing conditions. The fraction of carbohydrates in BSG was 25% in IRA and 35% in BSA and constituted 8% and 13% of monosaccharides, disaccharides and oligosaccharides. BSH contained 27% (BSH-IRA) and 40% (BSH-BSA) of carbohydrates, consisting mostly of monosaccharides, disaccharides and oligosaccharides; see Table 2.

Finally, as is well known, BSG is considered a rich source of dietary fiber. The BSG samples analyzed contained 39% (BSG-IRA) and 33% (BSG-BSA) of TDF. The TDF content of BSH samples (derived directly from hops), for both BSH-IRA and BSH-BSA, was around 12%; other authors in the literature reported slightly larger quantities (19–23%) [40].

Consuming food rich in dietary fiber has several important physiological effects on humans and has been associated with the prevention of several diseases. Dietary fiber consumption decreases the occurrence of cardiovascular diseases (hypertension, stroke, and coronary heart disease), diabetes, obesity, and gastrointestinal disorders, and also promotes normal bowel functioning and appears to improve immune function [41]. β-glucans and arabinoxylans are an important fraction of the fiber found in barley and have been implicated in lowering plasma cholesterol, improving lipid metabolism, and reducing glycemic index [42]. The β-glucans and arabinoxylans content in brewing by-products is reported in Figure 1. In BSG samples, the number of β-glucans was around 1% (0.84% *w*/*w* and 1.43% *w*/*w* for BSG-IRA and BSG-BSA, respectively); data fitted with results in literature [43]. The percentage of arabinoxylans in BSG, derived from the brewing process of barley malt beer and wheat beer, was 6.54% and 2.67%, respectively; other authors in the literature report higher concentrations (about 25%). In any case, the amount of arabinoxylan is greatly influenced by the raw materials used, the brewing process, and the BSG treatments (such as washes) [44]. The β-glucans were almost absent in BSH, while 1.37% and 1.46% of arabinoxylans were detected in both IRA- and BSA-BSH, respectively (probably composed of water-soluble arabinoxylans derived directly from the liquid trapped in the trub, which predominately consists of high-molecular weight protein) [31,45].

### 3.2. Total Polyphenols and Phenolic Acid Composition

Interesting concentrations of total polyphenols and phenolic acids were highlighted in both BSG and BSH samples. The amount of TP and PAs, as well as their content and profiles, are reported in Table 3. The content (dry basis) of TP in brewing spent grains was 7.30 mg GAE g^−1^ in the IRA sample and 9.55 mg GAE g^−1^ in BSA, and in both BSG samples mainly represented (about 70% of total polyphenols) by bound phenolic (BP) compounds. Other authors have also shown TP to be present in a similar, yet slightly lower, range [46,47]. The number of total polyphenols in brewing spent hop was 10.90 and 17.58 mg GAE g^−1^ in BSH-IRA and BSH-BSA, respectively. In both BSH samples, the TP fraction consisted mainly of free polyphenols (average of about 90% of TP).

For the phenolic acid fraction of BSG samples, the main representative PAs were: syringic, homovanillic, 3-hydroxybenzoic, vanillic, and sinapic acid for the free fraction of BSG-IRA. On the other hand, syringic, homovanillic, ferulic, and salicylic were main representative PAs for the free fraction of BSG-BSA. The PAs of the bound fraction were: ferulic, salicylic, *p*-coumaric, sinapic, caffeic, and syringic acid for both BSG-IRA and BSG-BSA. The values of total PA (TPA) were 1.5 mg g^−1^ in BSG-IRA and 2 mg g^−1^ in BSG-BSA, accounting for the main fraction (92–93%) of bound PAs (BPAs). Differences between the PA composition of BSG-IRA and BSG-BSA, mostly for the free fractions, were observed. These variations are probably due to the different raw materials employed and to the different brewing processes. The amount of TPA was lower in BSH if compared with BSG—0.055 mg g^−1^ in the IRA sample and 0.164 mg g^−1^ in BSA. The Ma sample showed a higher concentration of free PAs (FPAs) at 67%; on the other hand, a higher concentration of BPAs (76%) was observed in BSA trub. Even if the TP content in BSH was higher than that of BSG, the same was not found for the phenolic acids. Kobus-Cisowska et al. [40] reported on the presence of interesting amounts of flavonoids, catechin and epicatechin in hop, compounds that are not investigated in this research, but which are most likely also found among the phenolic compounds of BSH.

The two spent hop samples derived from two different ale beers differed in the profile and concentration of PAs. In addition, the difference was observed between the free (FP) and bound polyphenol (BP) fractions, but only for BSH of Imperial red ale beer. The major free PAs of the IRA sample were 3-hydroxybenzoic, homovanillic, salicylic, caffeic, 2,6-dihydroxybenzoic, and syringic acid. In turn, the dominant phenolic acids of the bound fraction were vanillic, syringic, homovanillic and ferulic acids. Both FP and BP fractions of BSH from the brewing of Belgian strong ale were characterized with the same phenolic acid profile; the major PAs were ferulic, salicylic, *p*-coumaric, and sinapic acids. The bound fraction contained *p*-coumaric, salicylic, ferulic, and sinapic acids. The differences between the composition of PAs in the two analyzed trubs probably depend on the different varieties and typologies of the hops used (aroma vs. bitter hops) and the different concentrations of hop. The total amount of hop products (extract + granules) in BSA beer was higher if compared to IRA beer (1.6 g/L versus 1.1 g/L). Moreover, in the recipe of BSA beer, 1.3 g/L of hop granules was used, while 39% less hop granules were used for the production of Ma beer (0.8 g/L). Considering that hop granules certainly bring more polyphenols than the higher apolar CO_2_ extracts, the higher number of phenolic compounds in BSH from brewing of BSA beer was easily predictable and justifiable.

### 3.3. Antioxidant Activity

In Figure 2, we report the results of the antioxidant activity of brewing by-products. The redox and radical systems used for antioxidant evaluation could influence the experimental results, and two or more redox and radical assays are necessary to investigate the radical scavenging and antioxidant capacities of selected samples. Thus, in the present work, three different assays (DPPH, ABTS and FRAP) were used for the evaluation of the antioxidant activity of the extracts obtained from BSG and BSH samples. The data show that all the extracts obtained from the brewing by-products have an interesting antioxidant activity. As known, the proper characteristics and composition, consisting mainly of the husk-pericarp-seed coat (where are contained most of the phenolic compounds of the barley grains), make BSG a valuable source of phenolic compounds and potentially bioactive material, the obtained results confirm this knowledge [48]. Moreover, the proper composition, in terms of the phenolic compounds of BSH, can explain the antioxidant potential underlined [49]. In particular, the results of three antioxidant assays underlined that the by-products collected during the brewing process of BSA beer showed the highest antioxidant potential (Figure 2). In particular, the BSG-IRA values of antioxidant activity (TE g^−1^ dm) were: 31.20 TE g^−1^ (dm) for FRAP, 15.31 TE g^−1^ (dm) for DPPH and 40.10 TE g^−1^ (dm) for ABTS, whereas the BSG-BSA values were: 42.13 TE g^−1^ (dm), 30.43 TE g^−1^ (dm) and 73.17 TE g^−1^ (dm), for the three assays, respectively. The antioxidant capacity of BSA spent grains, in terms of reducing potential, was 1.4-fold higher compared with the IRA sample, reaching values of 1.8-fold and 2-fold higher for scavenging activity. This behavior is probably due to the different raw materials used (in this regard, both the genotype and the cultivation conditions are relevant), the several modifications that occur during malting and roasting process (some studies have reported an increase in the antioxidant activity during malting, others have underlined the opposite effect) [50], as well as the conditions of each step of brewing process. Furthermore, even for BSH samples, higher antioxidant activities were observed in the samples collected from the BSA beer production process. The results, in TE g^−1^ (dm), for BSH-IRA were: 29.98, 28.62 and 63.24, for FRAP, DPPH and ABTS, respectively. The antioxidant values for BSH-BSA samples were: 35.75 TE g^−1^ (dm) for FRAP, 38.65 TE g^−1^ (dm) for DPPH and 78.05 TE g^−1^ (dm) for ABTS. The antioxidant results for BSH samples underlined the higher activity, even if the differences are less marked if compared with spent grains sample, of BSA spent hop. The values for BSA samples were between 1.2 and 1.4-fold higher than IRA. The highlighted behavior could be justified considering the different concentrations and typologies of the hop-products used during the production of BSA and IRA beers. As reported above for phenolic compounds, the recipe of BSA beer contained a larger amount of hop and in particular higher amount of hop granules (1.3 g/L versus 0.8 g/L) compared with IRA recipe; considering that hops antioxidant constituents are mainly phenolic compounds, the higher antioxidant capacity of BSH-BSA can be easily justify. Moreover, to better understand the antioxidant potential of brewing by-products, the amount of different brewing by-products required to give a 50% inhibition of the probe in each considered antioxidant assay (IC50) was also calculated [26]. The results are reported in Figure 3. The BSG-IRA values of IC50 were: 7.00 g/L for FRAP, 23.09 g/L for DPPH and 8.52 g/L for ABTS, while the BSG-BSA values were: 5.18 g/L, 11.61 g/L and 4.67 g/L. For BSG samples, the IC50 values of BSA samples were at least 1.4-fold (1.8-fold for ABTS and 2 for DPPH) higher in comparison with IRA. For BSH samples, the IC50 values for FRAP, DPPH and ABTS assays were: 7.28 g/L, 12.35 g/L and 5.40 g/L for BSH-IRA and 6.11 g/L, 9.14 g/L and 4.38 g/L for BSH-BSA. For BSH samples, the differences were less marked and the IC50 values for BSA samples were from 1.2 to 1.4-fold higher than IRA. These results underline the higher antioxidant power of by-products collected from the BSA beer production process. The strongest antioxidant activity was statistically confirmed. This result could be justified considering the higher content of phenolic compounds (both TP and TPA) in BSA samples. The values of the TP and the TPA in BSG-BSA sample were 1.3-fold higher in comparison with BSG-IRA; likewise, the contents of the TP and TPA of the BSH-BSA sample were 1.6 and 3 times higher in comparison with IRA spent hop. The obtained results for antioxidant capacity are positively correlated with the higher values of phenolic compounds (both TP and TPA) found in BSA by-products.

Other authors reported on the antioxidant activity of different extract of brewing by-products. Moreira et al. [51] reported on BSG microwave-assisted extracts; for the DPPH the absorbance decrease was measured after 120 min and the results were expressed as antiradical power (ARP% = 1/IC50 × 100), the ARP% values for different BSG analyzed ranged between 17% and 12%. The results obtained in this research, expressed as ARP% and referred to 1 g of samples (dry basis), were 4.33 and 8.61 for BSG-IRA and BSG-BSA, respectively. The ARP% values for BSH were 8.10 for IRA and 10.94 for BSA sample. Other authors [52] reported on the antiradical activity of BSG and BSH methanolic extracts, quantified by DPPH assay and expressed as inhibition %. The obtained results for their extract were 11.33% for BSG and 12.78% for BSH.

### 3.4. Validation of HPLC Method

The validation of an analytical method is important to judge the quality, reliability and relevance of the analytical results.

The parameters calculated for the system suitability assessment were the %RSD of the retention time and peak area for each PA, the number of theoretical plates (N), and the tailing factor (T). The results (reported in Table 1) for the %RSD of retention time and peak area, for the eighteen phenolic compounds, ranged between 0.3% and 1.4%. The %RSD values of N and T were between 0.3 and 1.5% and 0.2 and 1%, respectively. The results of the system suitability test were in the range of the acceptance criteria, emphasizing the suitability of the system [25].

Each phenolic compound was identified by its specific retention time and the fingerprint of UV absorbance at three different wavelengths (254, 278, and 324 nm), in comparison with the standards. The quantification was carried out using an external standard calibration. The linearity of the method is confirmed by the R values between 0.9965 and 0.9997 and the intercepts, which are very close to zero (Table 1). Cochran’s test was used to verify the homoscedasticity of the calibration plots. The assumption of homoscedasticity means that the standard deviations of the error terms are constant and do not depend on the x-value. The obtained Cc values for the calibration plots were lower than the value of C tabulated (Ct), as listed in the Cochran table for a level of significance of α = 0.01, indicating the homoscedasticity of each plot (Table 1).

Finally, in order to determine the efficiency of the whole procedure used for the quantification of free phenolic compounds in brewing by-products, the BSG samples were analyzed after spiking with different concentrations of PAs and tyrosol. Each step of the entire methodology (solid–liquid extraction and UHPLC–UV analysis) was replicated ten times.

The trueness of the method, expressed as the relative spike recovery Rʹ%, was calculated by analyzing fortified and unfortified samples ten times (*n* = 10) for four levels, ranging from 5 to 100 µg of eighteen PAs per g of BSG [28,29]. The results are reported in Table 4. The trueness values ranged between 80% and 110%. The obtained results are in the requested range when considering that the content of PAs in BSG and BSH is in the concentration range of µg/g and, according to the AOAC guidelines [47], the recovery limit for these concentrations it is expected to be between 80% and 115% [53]. The precision of the method was evaluated by calculating the relative standard deviations of repeatability (%RSDr) and the results are reported in Table 4. The obtained values ranged between 0.2 and 2.4 and are within the acceptability criteria for %RSD (±2%) [54].

The uncertainty of the developed method was tested calculating the Horwitz ratio value (HorRat); the obtained results are reported in Table 4. Under repeatability conditions, the accepted values are between 0.3 and 1.3. Lower values (˂0.3) may indicate excellent training and experience [30].

## 4. Conclusions

BSG and BSH are very attractive by-products for the improvement of the nutritional value of foods. They are a source of fiber and protein, which are staple nutritional components in the human diet. Consuming large amounts of food rich in dietary fiber has several important physiological effects in humans and has been associated with the prevention of several diseases. β-glucans and arabinoxylans, important fractions of the brewing fiber that are implicated in several nutritional benefits, were also found in BSG and BSH, contributing to their healthy value. The correct intake of dietary protein or a moderately greater intake than the Recommended Dietary Allowance (RDA) may be beneficial and may help to reduce the risk of chronic diseases such as obesity, cardiovascular disease, type 2 diabetes, osteoporosis, and sarcopenia [55]. Moreover, the high content of TP and, in particular, PA detected by a validated UHPLC–UV method, and the high antioxidant potential, verified by means of three different assays, confirm the health benefits of BSG and BSH residues from the brewing process. Moreover, the results of this study underline the potential interest in the health benefits provided by BSG obtained from the production of beer made from malt and wheat (mostly in relation to the claims made about BSG’s phenolic content and antioxidant properties). The influence of the amount, variety and type of hops used on the antioxidant potential of BSH was also highlighted. Owing to their valuable compositions, brewing by-products are interesting by-products to upcycle and use in the formulation of innovative functional foods (developed in order to control chronic bowel disorders and constipation, as well as cardiovascular diseases, in protein-rich diets, while maintaining muscle mass and strength as people age). Moreover, the well-known high value of these by-products in animal feedings (not only because of the reduced costs of disposal costs but also for the promising potential as functional ingredients) was also confirmed by their protein, fiber and carbohydrates concentration that make them a suitable supplement in ruminant diet [56] and also by the presence of phenols and antioxidant compounds that can considered functional ingredients, with beneficial anti-inflammatory and antimicrobial activity, in animal feed [57].

## Figures and Tables

**Figure 1 antioxidants-10-00165-f001:**
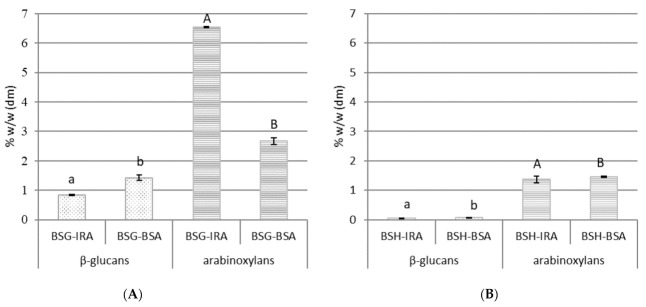
β-glucans and arabinoxylans content of the brewing spent grains, BSG (**A**), and spent hop, BSH (**B**) samples (% *w*/*w* dm); IRA = Imperial red ale beer brewing process; BSA = Belgian strong ale beer brewing process; *n* = 3; BSG = brewing spent grains; *w* = weight, *dm* = dry matter. Different letters in each column graph denote significant differences between samples (*p* ≤ 0.01).

**Figure 2 antioxidants-10-00165-f002:**
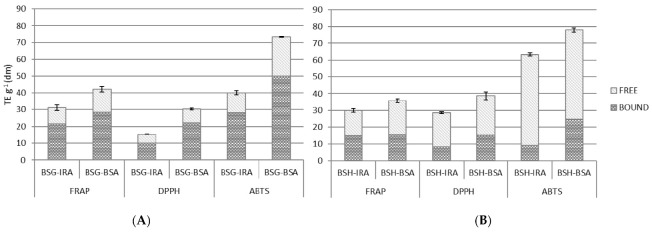
Antioxidant activities of the brewing spent grain, BSG (**A**) and brewing spent hop, BSH (**B**) samples (TE g^−1^ dm). *n* = 3; IRA = Imperial red ale; BSA = Belgian strong ale; TE = Trolox equivalent, dm = dry matter. Different letters and symbols in each column graph denote significant differences between samples (*p* ≤ 0.01).

**Figure 3 antioxidants-10-00165-f003:**
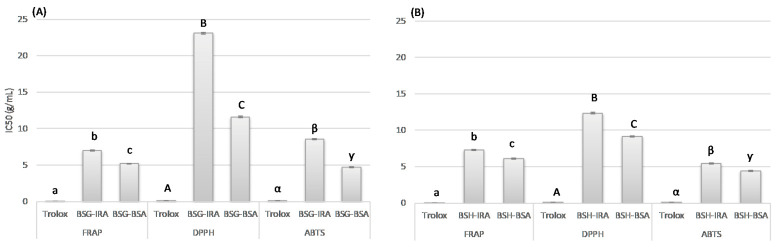
Antioxidant activities of the brewing spent grains, BSG (**A**) and brewing spent hop, BSH (**B**) samples (IC50); *n* = 3; IRA = Imperial red ale; BSA = Belgian strong ale; IC50 = The median inhibitory concentration. Different letters and symbols in each column graph denote significant differences between samples (*p* ≤ 0.01).

**Table 1 antioxidants-10-00165-t001:** Linearity and homoscedasticity test for calibration plots, results for system-suitability study, detection and quantitation limit values for phenolic compounds.

Phenolic Compounds	Cc	R	Linear Equation	t_R_ (min)		A		*N*		*T*		LOD	LOQ
				Mean	RSD%	Mean	RSD%	Mean	RSD%	Mean	RSD%		
3,4,5-trihydroxybenzoic	0.480	0.9997	y=357.723∗x−15.66	0.658	0.964	1724	0.616	2078	0.825	1.766	0.204	0.025	0.083
3,5-dihydroxybenzoic	0.401	0.9987	y=206.41∗x−12.274	1.199	0.772	999	0.325	3592	1.544	1.284	0.339	0.051	0.171
2,5-dihydroxybenzoic	0.401	0.9989	y=174.362∗x−1.260	1.774	1.392	876	0.786	5072	1.608	1.417	1.153	0.039	0.129
4-hydroxybenzoic	0.327	0.9977	y=779.903∗x−1.410	2.326	1.250	3845	0.680	8664	1.521	1.289	0.198	0.049	0.165
tyrosol	0.446	0.9979	y=64.502∗x−0.281	2.577	0.989	317	0.855	4773	1.469	1.110	0.749	0.041	0.136
2,6-dihydroxybenzoic	0.578	0.9988	y=174.789∗x−64.544	2.973	0.804	700	0.876	4380	1.310	1.401	0.442	0.036	0.122
3-hydroxybenzoic	0.502	0.9989	y=49.130∗x	3.426	0.518	243	1.336	11,089	0.816	0.901	0.273	0.071	0.236
vanillic	0.342	0.9993	y=365.689∗x	4.137	0.471	1805	0.681	16,231	0.751	1.019	1.078	0.038	0.126
caffeic	0.571	0.9988	y=481.178∗x	4.713	0.577	2356	1.135	21,041	1.202	0.769	0.691	0.066	0.218
chlorogenic	0.475	0.9975	y=413.348∗x−38.745	5.061	0.519	1852	1.390	24,311	0.644	1.329	0.125	0.050	0.167
homovanillic	0.398	0.9965	y=55.192∗x−12.406	5.749	0.583	249	0.945	88,626	0.619	1.020	0.412	0.017	0.058
syringic	0.400	0.9987	y=321.769∗x−0.275	5.941	0.771	1592	0.647	53,020	0.854	0.695	0.271	0.027	0.089
*p*-coumaric	0.504	0.9977	y=658.517∗x	6.983	0.433	3170	0.836	122,209	0.262	1.465	0.146	0.051	0.167
salicylic	0.457	0.9971	y=42.960∗x	7.471	0.594	206	6.554	62,152	1.043	0.570	0.797	0.047	0.156
ferulic	0.462	0.9971	y=670.768∗x	7.799	0.499	3207	0.299	152,478	0.482	1.569	0.095	0.037	0.124
*m*-coumaric	0.467	0.9978	y=587.344∗x	8.026	0.431	2817	0.366	102,787	0.256	1.243	0.141	0.060	0.199
sinapic	0.504	0.9997	y=227.972∗x	8.211	0.300	1119	0.727	168,685	0.571	1.323	0.853	0.046	0.153
*o*-coumaric	0.434	0.9990	y=593.968∗x	9.161	1.007	2601	0.203	211,161	0.197	1.359	0.117	0.026	0.086

Mean of ten replications. Cc: Cochran’s constant calculated; R, correlation coefficient; t_R_, retention time; A, peak area; RSD, relative standard deviation; *N*, number of theoretical plates; *T*, tailing factor; LOD, limit of detection; LOQ, limit of quantitation.

**Table 2 antioxidants-10-00165-t002:** Composition of spent grains (BSG) and spent hop (BSH) from the brewing of Imperial red ale (IRA) and Belgian strong ale (BSA) beers.

		Brewing Spent Grains		Brewing Spent Hop	
		IRA	BSA	IRA	BSA
Moisture% (fresh product)		78.69 ± 0.92 ^a^	80.49 ± 0.36 ^b^	83.83 ± 0.37 ^A^	79.18 ± 0.86 ^B^
Moisture% (dried product)		8.61 ± 0.70 ^a^	7.07 ± 0.07 ^b^	5.30 ± 0.15 ^A^	5.41 ± 0.02 ^A^
Water activity (a_w_)		0.41 ± 0.01 ^a^	0.43 ± 0.01 ^b^	0.35 ± 0.01 ^A^	0.42 ± 0.01 ^B^
Total nitrogen (% dm)		2.97 ± 0.01 ^a^	2.55 ± 0.01 ^b^	8.33 ± 0.16 ^A^	6.35 ± 0.02 ^B^
Proteins (% dm)		18.58 ± 0.02 ^a^	15.91 ± 0.05 ^b^	52.02 ± 1.03 ^A^	39.67 ± 0.03 ^B^
Ash (% dm)		2.96 ± 0.10 ^a^	2.63 ± 0.0.05 ^b^	2.33 ± 0.07 ^A^	2.11 ± 0.08 ^B^
Fat (% dm)		6.75 ± 0.16 ^a^	6.40 ± 0.09 ^b^	1.23 ± 0.02 ^A^	1.06 ± 0.01 ^A^
Total dietary fiber (% dm)		38.52 ± 0.12 ^a^	33.01 ± 2.76 ^b^	12.38 ± 1.13 ^A^	12.19 ± 0.73 ^A^
Carbohydrates (% dm)		24.58 ± 0.71 ^a^	34.97 ± 1.94 ^b^	26.78 ± 1.75 ^A^	39.94 ± 0.91 ^B^
Sugars (g 100^−1^ g dm)	Fructose	0.30 ± 0.01 ^a^	0.37 ± 0.01 ^b^	0.49 ± 0.00 ^A^	0.89 ± 0.00 ^B^
	Glucose	1.33 ± 0.02 ^a^	1.81 ± 0.01 ^b^	1.67 ± 0.00 ^A^	2.79 ± 0.04 ^B^
	Sucrose	0.14 ± 0.01 ^a^	0.29 ± 0.00 ^b^	7.62 ± 0.05 ^A^	11.78 ± 0.00 ^B^
	Maltose	3.19 ± 0.02 ^a^	5.94 ± 0.03 ^b^	9.06 ± 0.05 ^A^	13.59 ± 0.14 ^B^
	D3 (maltotriose)	1.13 ± 0.01 ^a^	1.98 ± 0.01 ^b^	2.41 ± 0.10 ^A^	3.99 ± 0.01 ^B^
	D4 (maltotetraose)	0.77 ± 0.04 ^a^	1.83 ± 0.06 ^b^	0.81 ± 0.06 ^A^	1.11 ± 0.05 ^B^
	D5 (maltopentaose)	0.18 ± 0.01 ^a^	0.24 ± 0.03 ^b^	0.48 ± 0.01 ^A^	0.42 ± 0.03 ^B^
	D6 (maltohexaose)	0.22 ± 0.01 ^a^	0.53 ± 0.05 ^b^	0.53 ± 0.05 ^A^	0.59 ± 0.16 ^A^
	D7 (maltoeptaose)	0.15 ± 0.00 ^a^	0.19 ± 0.00 ^b^	0.41 ± 0.07 ^A^	0.25 ± 0.01 ^B^
	D8	0.02 ± 0.00 ^a^	0.04 ± 0.01 ^b^	0.10 ± 0.01 ^A^	0.14 ± 0.01 ^B^
	D9	0.02 ± 0.00 ^a^	0.04 ± 0.00 ^b^	0.06 ± 0.00 ^A^	0.08 ± 0.01 ^B^
	D10	0.02 ± 0.00 ^a^	0.03 ± 0.00 ^b^	0.03 ± 0.00 ^A^	0.04 ± 0.00 ^B^
	D11	0.01 ± 0.00 ^a^	0.02 ± 0.00 ^b^	0.02 ± 0.00 ^A^	0.04 ± 0.00 ^B^
	D12	0.02 ± 0.00 ^a^	0.02 ± 0.00 ^a^	0.02 ± 0.00 ^A^	0.04 ± 0.01 ^B^
	D13	0.01 ± 0.00 ^a^	0.01 ± 0.00 ^b^	0.03 ± 0.00 ^A^	0.02 ± 0.00 ^B^
	D14	0.01 ± 0.00 ^a^	0.01 ± 0.00 ^a^	0.02 ± 0.00 ^A^	0.03 ± 0.00 ^B^
	D15	-	-	0.01 ± 0.00	-
	Total sugars	7.51 ± 0.05 ^a^	13.33 ± 0.00 ^b^	23.76 ± 0.27 ^A^	35.78 ± 0.11 ^B^

*n* = 3; Values in the same row followed by different lowercase letters for BSG, and uppercase letters for BSH, are statistically different, *p* ≤ 0.01; IRA = Imperial red ale, BSA = Belgian strong ale, dm = dry matter.

**Table 3 antioxidants-10-00165-t003:** Free and bound polyphenols content (mg GAE g^−1^ dm) and phenolic acids profile (µg g^−1^ dm) of spent grains and spent hop samples from the brewing of Imperial red and Belgian strong ale beers.

	Brewing Spent Grains		Brewing Spent Hop	
	IRA	BSA	IRA	BSA
FP (mg GAE g^−1^ dm)	2.49 ± 0.12 ^a^	2.44 ± 0.12 ^a^	9.51 ± 0.11 ^A^	15.73 ± 1.30 ^B^
BP	4.80 ± 0.11 ^a^	7.11 ± 0.10 ^b^	1.38 ± 0.12 ^A^	1.84 ± 0.02 ^B^
TP	7.30 ± 0.23 ^a^	9.55 ± 0.22 ^b^	10.90 ± 0.23 ^A^	17.58 ± 1.32 ^B^
FPA (µg g^−1^ dm)				
4-hydroxybenzoic	3.89 ± 0.05 ^a^	6.89 ± 0.06 ^b^	0.77 ± 0.05	-
2,6-dihydroxybenzoic	-	-	3.09 ± 0.01	-
Vanillic	14.63 ± 0.06 ^a^	5.40 ± 0.02 ^b^	1.41 ± 0.01	-
3-hydroxybenzoic	19.74 ± 0.75	-	9.59 ± 0.34	-
Caffeic	1.87 ± 0.09	-	6.19 ± 0.03	-
Chlorogenic	-	1.31 ± 0.03	0.77 ± 0.01	-
Syringic	34.33 ± 0.16 ^a^	67.51 ± 0.31 ^b^	3.02 ± 0.16	-
Homovanillic	19.99 ± 0.84 ^a^	39.32 ± 1.66 ^b^	9.16 ± 0.14	-
*p*-coumaric	-	4.44 ± 0.06	0.76 ± 0.02 ^A^	2.20 ± 0.22 ^B^
Salicylic	-	8.48 ± 0.31	6.12 ± 0.44 ^A^	16.49 ± 0.92 ^B^
Ferulic	3.96 ± 0.27 ^a^	15.53 ± 0.60 ^b^	0.38 ± 0.02 ^A^	14.97 ± 1.40 ^B^
*m*-coumaric	-	1.64 ± 0.06	-	-
Sinapic	10.96 ± 0.85 ^a^	3.79 ± 0.12 ^b^	1.01 ± 0.05 ^A^	6.77 ± 0.19 ^B^
Total FPA	109.36 ± 3.08 ^a^	158.86 ± 3.20 ^b^	36.96 ± 1.25 ^A^	40.43 ± 2.73 ^B^
BPA (µg g^−1^ dm)				
3,5-dihydroxybenzoic acid	-	-	-	6.73 ± 0.03
2,5-dihydroxybenzoic acid	-	-	-	4.70 ± 0.26
4-hydroxybenzoic	4.11 ± 0.26 ^a^	12.73 ± 0.10 ^b^	0.55 ± 0.03	-
Vanillic	11.46 ± 0.40 ^a^	1.28 ± 0.01 ^b^	38.75 ± 1.27	-
Caffeic	39.78 ± 0.70 ^a^	89.68 ± 2.65 ^b^	-	1.31 ± 0.07
Chlorogenic	-	-	1.07 ± 0.01 ^A^	0.89 ± 0.010 ^B^
Syringic	27.36 ± 3.00 ^a^	19.52 ± 0.16 ^b^	8.21 ± 0.74	-
Homovanillic	12.12 ± 0.96 ^a^	9.36 ± 0.20 ^b^	2.65 ± 0.08	-
*p*-coumaric	256.38 ± 4.09 ^a^	584.84 ± 5.17 ^b^	-	18.68 ± 0.12
Salicylic	491.56 ± 30.99 ^a^	102.36 ± 0.73 ^b^	1.80 ± 0.01 ^A^	28.24 ± 0.15 ^B^
Ferulic	476.98 ± 11.82 ^a^	692.15 ± 6.81 ^b^	2.56 ± 03.05 ^A^	51.72 ± 0.19 ^B^
Sinapic	116.25 ± 4.51 ^a^	322.97 ± 20.85 ^b^	-	10.72 ± 0.22
Total BPA	1431.87 ± 56.47 ^a^	1859.60 ± 37.85 ^b^	17.94 ± 0.92 ^A^	123.17 ± 1.04 ^B^
TPA	1541.24 ± 59.55 ^a^	2018.46 ± 41.05 ^b^	54.90 ± 2.17 ^A^	163.60 ± 3.78 ^B^

*n* = 3; Values in the same row followed by different lowercase letters for BSG, and uppercase letters for BSH, are statistically different, *p* ≤ 0.01; IRA = Imperial red ale; BSA = Belgian strong ale; FP = free polyphenols; BP = bound polyphenols; TP = total polyphenols; FPA = free phenolic acids; BPA = bound phenolic acids; TPA = total phenolic acids; dm = dry matter.

**Table 4 antioxidants-10-00165-t004:** Trueness (R’%) and precision (RSDr%) and Horwitz ratio value (HorRat) of the optimized method, based on four concentration levels (5, 25, 50, and 100 µg/g) of phenolic compounds spiked brewing spent grains.

Phenolic Compounds	Concentration Level (µg/g)											
	5			25			50			100		
	R’% (Mean ± SD)	RSDr %	HorRat	R’% (Mean ± SD)	RSDr %	HorRat	R’% (Mean ± SD)	RSDr %	HorRat	R’% (Mean ± SD)	RSDr %	HorRat
3,4,5-trihydroxybenzoic	100 ± 1	0.9	0.1	101 ± 1	0.2	0.1	98 ± 2	1.0	0.1	93 ± 2	1.6	0.2
3,5-dihydroxybenzoic	85 ± 1	0.8	0.1	98 ± 1	0.7	0.1	96 ± 2	1.0	0.1	90 ± 1	0.5	0.1
2,5-dihydroxybenzoic	87 ± 1	1.3	0.1	89 ± 1	1.1	0.1	81 ± 4	1.2	0.4	85 ± 2	1.2	0.1
4- hydroxybenzoic	85 ± 1	1.1	0.1	92 ± 1	0.6	0.1	86 ± 2	1.2	0.1	82 ± 1	1.1	0.1
tyrosol	93 ± 1	0.9	0.1	86 ± 2	2.3	0.2	89 ± 4	2.1	0.2	94 ± 1	0.9	0.1
2,6-dihydroxybenzoic	92 ± 1	1.5	0.1	90 ± 1	0.9	0.1	97 ± 3	1.7	0.1	84 ± 1	0.8	0.1
3- hydroxybenzoic	103 ± 1	1.0	0.1	101 ± 1	1.2	0.1	109 ± 4	1.9	0.2	102 ± 1	1.1	0.1
vanillic	88 ± 1	0.8	0.1	92 ± 2	1.7	0.1	90 ± 4	2.2	0.1	80 ± 1	0.3	0.1
caffeic	101 ± 1	1.1	0.1	100 ± 1	0.8	0.1	110 ± 3	1.2	0.1	104 ± 1	1.1	0.1
chlorogenic	97 ± 1	1.2	0.1	85 ± 2	2.1	0.1	93 ± 2	2.4	0.1	81 ± 1	1.0	0.1
syringic	88 ± 1	1.2	0.1	93 ± 2	1.6	0.1	89 ± 3	1.0	0.1	80 ± 1	0.6	0.1
homovanillic	107 ± 1	0.6	0.1	87 ± 1	1.5	0.1	86 ± 2	1.2	0.1	80 ± 1	0.2	0.1
*p*-coumaric	94 ± 1	1.4	0.1	86 ± 1	1.0	0.1	94 ± 3	1.6	0.1	81 ± 2	1.1	0.2
salicylic	90 ± 1	1.1	0.1	82 ± 1	1.4	0.1	93 ± 1	0.6	0.1	86 ± 1	1.1	0.1
ferulic	97 ± 1	2.0	0.1	92 ± 1	0.6	0.1	85 ± 3	1.9	0.2	80 ± 1	0.5	0,1
*m*-coumaric	97 ± 1	1.0	0.1	83 ± 2	2.3	0.2	84 ± 3	1.6	0.1	85 ± 1	1.1	0.1
sinapic	96 ± 1	1.5	0.1	80 ± 1	1.6	0.1	80 ± 1	0.4	0.1	84 ± 1	1.1	0.1
*o*-coumaric	104 ± 1	1.7	0.1	103 ± 1	0.2	0.1	93 ± 2	0.9	0.1	81 ± 1	1.2	0.1

Mean of ten replications. R’ = relative spike recovery; RSDr = relative standard deviations of repeatability; HorRat = Horwitz ratio value.

## Data Availability

The data presented in this study is available in the article.

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
