# Peer review of "Brewing By-Product Upcycling Potential: Nutritionally Valuable Compounds and Antioxidant Activity Evaluation"

_antioxidants, 2021, doi:10.3390/antiox10020165_

Round 1

Reviewer 1 Report

The manuscript with the title ‘Brewing by-products upcycling potential: nutritional valuable compounds and antioxidant activity evaluation' is interesting and fits the scope of Antioxidants journal. However, I found several weak points that should be addressed before considering for publication. My comments and concerns are listed below, in the hope they will help to improve the scientific quality of this manuscript.

GENERAL COMMENTS:

  1. I suggest the revision of the manuscript by a native speaker.
  2. All abbreviations in the text, tables, and figures should be explained in their first appearance.
  3. I suggest include in the manuscript a figure showing representative chromatograms of BSG and BSH extracts with marked quantified phenolics.

More specific comments are included in the attached pdf file.

Author Response

Manuscript ID, antioxidants-1026061

Brewing by-products upcycling potential: nutritional valuable compounds and antioxidant activity evaluation

REVIEWER 1

Dear Referee,

The suggested revisions are made considering your precious comments.

The manuscript with the title ‘Brewing by-products upcycling potential: nutritional valuable compounds and antioxidant activity evaluation' is interesting and fits the scope of Antioxidants journal. However, I found several weak points that should be addressed before considering for publication. My comments and concerns are listed below, in the hope they will help to improve the scientific quality of this manuscript.

GENERAL COMMENTS:

  1. I suggest the revision of the manuscript by a native speaker.
  2. All abbreviations in the text, tables, and figures should be explained in their first appearance.
  3. I suggest include in the manuscript a figure showing representative chromatograms of BSG and BSH extracts with marked quantified phenolics.

More specific comments are included in the attached pdf file.

  • Ok, the paper has now been revised by MDPI English-editing service
  • Ok, the text was checked and corrected
  • Chromatograms of BSG and BSH are not so directly representative of effective concentrations of PAs in brewing by-products samples because more dilutions of the same extract, obtained from 1g of sample, were injected into UHPLC system to obtain the presented results, in fact each phenolic compound give a different response in UV, some are more sensitive some other less. The authors think that include the cromatograms with marked amounts is not useful. Anyway, if the referee wants its, the authors  can include the chromatograms with the name of the PAs (for example MaWhBSG and MaWhBSH free Pas, or others)

The other response to referee comments are reported below

  1. Please give more precise information on results and sample comparisons.

Lines 27-30

More information has been added along the text.

  1. controversial statement about brewing by-products. according to EU legislation the terms 'by-product' and 'waste' are probably not synonymous

Ok. The authors checked the text, and made corrections when mistakenly the term waste was used instead of  by-products

  1. Consider changing to phenolics or polyphenols

The keyword was changed in phenolics

  1. Consider adding information on EU legislation concerning disposal/re-use of brewing by-product.

Lines 68-73 .The EU instructions on disposal of brewing by-products were added

  1. Authors give several examples of possible use of brewing by-product, but I am also interested in a precise information how this material is currently used in EU. Giving specific numbers will emphasize the scale of the problem and the importance of the issue undertaken by the authors.

Lines 68-73 .The EU instructions on disposal of brewing by-products were added

  1. Chemicals section is missing. Reagents, standards, etc.?

OK, the chemicals section was add.

  1. For me it is better to describe it as 'barley malt beer' not 'all malt' because malt can come from barley or wheat or other.

and For me it is better to describe it as 'barley malt + wheat beer', because still barley malt is major constituent.

OK, the descriptions have been modified along the entire manuscript

  1. Were samples taken from one batch or more?

The samples were blend of three different batches, the information was added along the manuscript lines  116-120

  1. What this parameter indicates, the author probably did not comment on more broadly in the results and discussion?

The water activity was measured to have a complete analytical characterization of the considered raw materials and because the authors wanted to have an idea of the stability of the freeze-dried and dried product. The value is shown in table 1, and now commented along the text lines 285-289

  1. I do not see this paper in References.

The reference was added in references section line 166

  1. or dissolved? Suspension is not adequate state for spectrophotometric measurements..

The observation is correct, the text was revised line  172

  1. 70% MeOH was used for extraction of plant material, and after evaporation of solvent the authors redissolved it in 1 mL without using any organic solvent, such as methanol? As far as I know, phenolic acids are hardly soluble in water without adding the organic solvent, please check it.

Yes there was a mistake, the solution for redissolved phenolic acids was 30% of methanol and 70% phase A, the information was add in the text lines 172 and 183

  1. methanolic extracts from brewing by-products

The authors reported the antioxidant power as uM TE per g of by-products, so they evaluated the antioxidant activity of the brewing by-product itself (BSG and BSH) and not only of the extracts (uM TE per g DW)

  1. Why after 24 h? in general the measurement is after 30 min incubation, because reaction is very fast.

Sorry. There was a mistake the exact timing of reaction was 30 min, the text has been  corrected. Line 201

  1. per L of extract

or

per g of sample DM?

Sorry. There was a mistake the exact unit of measurement was μmol Trolox equivalents (TE) g−1 (dry basis) of sample. Line 204

  1. I miss the information how standards were prepared (stock and working solutions).

Quantitation was made at 3 different UV wavelengths but it is not specified among acids.

The manuscript has been revised, lines 217-224

A  stock solution of 100ug/ml of a mix of considered phenolic acids was used to prepare working solutions and  the calibration curve was plotted at the best wavelength for each phenolic compounds considering their absorption spectrum.

  1. Please check the correctness of given range. According to 2.2.7 subsection the samples' concentration for HPLC analysis was 1 g DM/mL, and looking at obtained results (up to 700 ug/g) and spike levels (up to 100 ug/g) it is hard to believe that given range was adequate for quantitation.

The range is correct (the calibration curves were constructed between 0.5 and 5 ug/ml)

The sample weighted for the extraction was 1g, of freeze dried or dried sample, after the extraction the samples and the spiked samples were diluted at different levels of dilution to reach the right concentration to fit the range of concentration of calibration plots. Moreover each phenolic acid has a different response if compared to the others and so different dilution were done to reach the correct concentration to be included in the calibration plot and to be quantified.

  1. The usually given parameter for linear regression model is coefficient of determination (R2) together with linear equation.

The correlation coefficient is also used to verify the linearity of a calibration plot, as reported by Ravisankar et al (in the references section) "The linearity of a method is a measure of how well a calibration plot of response vs. concentration approximates a straight line....The resulting plot slope, intercept and correlation coefficient provide the desired information on linearity."

The value reported in table 3 are the correlation factor of the plot for each PA and tyrosl, so they are the R and not the R2.

The program that we used to calibrate the instrument reported the correlation factor too. Anyway, if the author prefer we can add the R2 to the table.

  1. Please specify the type of fortified samples - raw material before extraction or obtained extracts.

The authors spiked the raw materials and not the extracts, with samples we talk about dried BSG (line 248  the word BSG was added along the text)

  1. In my opinion, if possible the full names of column titles, headings should be used instead abbreviations - for clarity reason and to limit footnotes.

See also other Tables.

Table 1. Thank you, the authors revised all the tables in the manuscript and full names were added in place of abbreviations

  1. In my opinion two decimal places is adequate precision, as above.

Table 1. Thank you for the observation, if the authors will delete the third decimal places some SD  of some values would become 0.00 instead of 0.001 so we preferred three decimal place, however if the referee prefers we can remove a decimal place

  1. lack of superscript letters

Table 1. Thank you for the observation, the superscript letters were added

  1. This abbreviation is not present, there is MaWh. Check also other tables and text.

Table 1. Thank you, there was a mistake, the abbreviation was revised and the manuscript and all the tables checked

  1. For me, the type of grain for malt is rather external factor.

Ok, to avoid any mistake the author rephrased the sentence eliminating the distinction between intrinsic and extrinsic factors (lines 287-289)

  1. The sentence does not compare the authors' results with the cited publication.

Ok, the text was improved and a comment about the results of cited authors was reported along the text (line 308-312)

  1. The sentence does not compare the authors' results with the cited publication.

Ok, a comment about the results of cited reference was reported along the text (lines 323-326)

  1. did not find info on how TP values were calculated (sum of FP and BP?).

Ok, thank you for your observation. Yes the TP were calculated as the sum of free and bound polyphenols and has been reported along the text in TP  first appearance. lines 130-131.

  1. If the authors knew/presented the differences in wort hopping of these two types of beer (specific values, type of material - cones, granules, extract), more interesting conclusions could be presented referring to this.

Ok, thank you for your observation. The commercial brewery that provided the brewing by-products analyzed gave to the authors only information on the quantities used per liter of beer and the type of hops used. The authors added these information along the manuscript to enrich the discussion: in materials and method section (lines 114-120), in results and discussion section, subsection 3.2 phenolic compounds and antioxidant activity, lines 377-382

  1. The authors' analysis of the results is very cursory, I encourage authors to analyze it in more depth. I also suggest comparing the results for the extracts with the positive control - Trolox, by calculating for example the IC50 values for all. Then it can be deducted about high, medium or low activity.

The authors also did not refer to the results of other researchers.

Thank you for the observation, the authors have enriched the discussion and calculated the IC50 for the by-products samples to make more evident the antioxidant activity showed for considered samples

Other authors reported on the antioxidant activity of different extract of brewing by-products, anyway the results are not directly comparable because the extraction procedure and  the measurements units are different, anyway the data give an idea of the antioxidant activity measured by other authors before. The authors reported these references along the manuscript

  1. or R2?

R value

As mentioned above the authors talk about R and not R2, anyway if the referee prefers R2 the authors can easily calculate the R2 and reported the values in the table 3

  1. Linear equations can be included in the manuscript.

Linear equations have been reported in table 3

  1. ? this compound is not acid

Right, tyrosol is an alcoholic compound, the text and the tables were revised distinguishing tyrosol from the other PAs

  1. It is not specified, the efficiency of which type of extraction used - free phenolics or alkaline hydrolysis, was determined with sample spiking. Sample fortifing with free forms of phenolics is probably only adequate for method of determination of free phenolics but not bound forms.

Right, the addition of  free PAs an tyrosol  was used to verify the efficiency of the extraction method of FPAs, it was specified and clarified along the text (line 437)

  1. add detector

The UV detector was added  along the text (line 470)

  1. The fragment of the discussion on this issue is poorly developed and not very specific.

As reported above, the authors added information and discussion about the hop products and their influence on antioxidant activity of BSH  and in results and discussion section, subsection 3.2 phenolic compounds and antioxidant activity

  1. What about feed purpose?

The authors added the purpose of formulation of functional foods with the BSG and BSH (or their active extracts) as raw materials. Lines 477-479

EXTRA REVIEWING

The authors have found an error in table 2. The reported values for total BPA and, as consequence, TPA value were wrong, the authors now reported the exact values in table 2 and along the text line 359.

Reviewer 2 Report

see attached.

Author Response

Manuscript ID, antioxidants-1026061

Brewing by-products upcycling potential: nutritional valuable compounds and antioxidant activity evaluation

REVIEWER 2

Explanations to the Referee

Dear Referee,

The suggested revisions are made considering your precious comments (our comments are underlined).

Brief summary/generic comments:

The authors tested brewing by-products in terms of their composition and antioxidant activity

to gain more information about their potential reusage in other products. The study is well

designed, and the main focus lies on the analytics. However, there is one main issues:

 How was the sampling done; i.e. how was it assured that the spent grain sample and the hot trub

sample were homogenous? Particularly the spent grain contains different layers with a very

inhomogeneous composition.

The BSG were collected immediately after lautering (blend of three different batches) and the BSH after wort boiling (blend of three different batches). Both BSG and BSH samples were carefully mixed before the sampling to avoid inhomogeneity. Lines 116-120

Further points to be revised:

Page 1, introduction: Please add the information that brewers spent grain can also be used

for biogas production.

Thank you for you precious suggestion, the information was added along the text and the relative reference added in reference section lines 64-65 ref 8

Page 2/3, lines 93-95: Please state which hop products were used. This is important

information as hop extracts contain no to little concentrations of phenolics while hop pellets

contain higher concentrations. 

Thank you for your comment, more information about hop products and relative comment were added along the text lines 114-117

Page 3, lines 97-102: As stated before, there is no information given on how the by-products

were homogenized during sampling. Both products are inconsistent in their composition and

depending on the sampling, great differences in their composition can be found. Please add

this information and discuss in your results section.

The information was added along the text Lines 115-120

Page 3, 2.2.3: This procedure of simply giving carbohydrates as the difference when

subtracting all other analyzed parameters is too vague. I would recommend omitting giving

this value as also the concentrations given in table 1 seem to be very high.

The procedure of giving carbohydrates as the difference when subtracting all other analysed parameters is often used to calculate carbohydrates in food samples reported also in FAO Food and nutrition technical paper77 (report of technical workshop of Food and Agriculture Organization of the United Nations, Rome 2003). The authors think that this is an important parameter for the characterization of BSG and BSH in the optic of their use as ingredient in functional food recipe. Anyway, if the referee prefers that these values are omitted we will delete these from the manuscript.

Page 7, lines 260-265: Please discuss the carbohydrate content in greater detail by including

literature and comparing your data to literature data (i.e.

https://onlinelibrary.wiley.com/doi/pdf/10.1002/j.2050-0416.2008.tb00774.x)

The discussion was implemented and data compared with literature. Lines 308-312

Page 6, lines 190-219: This section and its results can be omitted or moved to

supplementary data as it is ancillary and not the central theme of this study. It may be only

relevant for readers who are interested in this extra information.

The authors think that the validation procedure of the UHPLC method, validated for analysing the samples of the present research and used for obtain the results showed  in the paper, is an important section of the research project and an important tool to ensure the reliability of the scientific data. So the author would prefer to leave the validation section in the present scientific paper.

EXTRA REVIEWING

The authors have found an error in table 2. The reported values for total BPA and, as consequence, TPA value were wrong, the authors now reported the exact values in table 2 and along the text line 364

Round 2

Reviewer 1 Report

The second version of the manuscript is improved, however, in my opinion, there are still several weak points that should be answered and clarified by the authors.

General comments:

I. Some results and discussion are only briefly described without deeper analysis.

II. I have a subjective feeling that the authors' results (chemical analysis of brewing by-products and antioxidant activity using very simple in vitro tests) do not bring much new information to what is already known about the nutritional potential of brewing spent grain and hops. Several good publications on the nutritional characteristics and possible uses of BSG and BSH are available. The authors do not propose any new possibilities of using this by-product. The decision in this matter obviously belongs to the editor.

More specific comments are included in the attached pdf file. I marked comments with numbers 1-27. Please, treat other my modifications of the manuscript only as suggestions directed toward its scientific improvement.

Author Response

Dear Referee 1

Thank you for your contribution, below you will find the answers to each observation

The second version of the manuscript is improved, however, in my opinion, there are still several weak points that should be answered and clarified by the authors.

General comments:

  1. Some results and discussion are only briefly described without deeper analysis.

All right. The results were deeper analyzed and the discussion improved

  1. I have a subjective feeling that the authors' results (chemical analysis of brewing by-products and antioxidant activity using very simple in vitro tests) do not bring much new information to what is already known about the nutritional potential of brewing spent grain and hops. Several good publications on the nutritional characteristics and possible uses of BSG and BSH are available. The authors do not propose any new possibilities of using this by-product. The decision in this matter obviously belongs to the editor.

The authors disagree with this observation, several valid research papers are available for BSG, but at our knowledge very poor researches were done on BSH and in particular on its antioxidant activity and composition, mostly in term of PAs; the authors think that their research can contribute to fill these gaps and to enrich the entire knowledge on a such current and interesting theme in the optic of upcycling and of a zero waste society

More specific comments are included in the attached pdf file. I marked comments with numbers 1-27. Please, treat other my modifications of the manuscript only as suggestions directed toward its scientific improvement.

The other modifications suggested were evaluated and the entire manuscript revised on the bases of those observations

  1. In my opinion, comparing the results from different batches would also be of great value because would show the true variability of the material studied, not just the varability of analytical methods. Why authors omitted this aspect of the study?

The three different batches (for each considered beer) supplied by the brewing industry (micro-brewing process) were made with the same raw materials (barley malt, wheat, different typology of hop) so the only variables that distinguish a batch from another were the process parameters, little differences or error of process, that were avoid using a blend of three different batches made with the same raw materials. The authors chosen to blend three different batches made with the same raw materials (of the same batch, microbreweries use the same batch of raw materials to make several batches) to cancel variables and any process errors in order to characterize the by-products only on the bases of the recipe and the raw materials used.

  1. Actually, this is more securing or some kind of sample processing than sample preparation.

Ok, the text has been revised

  1. The source (article) of this factor should be added.

The reference was added along the text and in the references section

  1. According to authors' answer to one of my previous comments, the samples were appropriately diluted before HPLC analyzes, but there is no information how much samples were dilluted and which solvent was used. Please, consider adding.

After the last step of extraction procedure, the solvent was evaporated to dryness and the extract redissolved into 1 mL of 30% methanol in eluent A.

In the following steps, the dilution (when necessary) was in eluent A and the dilution was at different volumes as a function of PAs concentration in the extracts, sometimes the dilution was 1 to 4 some other 1 to 8 and so on as needed. The information was added along the text (lines 179-180)

  1. Studies of various research groups indicate that better results and efficiency of phenolic acid extraction are obtained with the use of ascorbic acid and in the dark, did the authors use such conditions, if not why?

The authors based their extraction on the method of Stagnari et al, the entire extraction was conducted in the dark and the samples were kept in the ice to avoid heating and loss of phenolic compounds. The information was added along the text (lines 252-253). No antioxidants were added during the extraction. The authors considered that the measures taken were enough to obtain a valid yield of phenolic compounds (valid yield confirmed also by Recovery values) .

  1. According to authors' answer to one of my previous comments, the samples were appropriately diluted before HPLC analyzes, but there is no information how much samples were dilluted and which solvent was used.

As written above (point 4), the dilution was in eluent A and the final volume of diluted sample was different in function of PAs concentration in the extracts

  1. Also information on the calculation of IC50 values should be added here.

Ok, the information was added along the text (lines 208-209)

  1. Add solvent which was used to prepare stock and working solutions of standards.

Ok, the information was added along the text (line 223)

  1. The UV detectors are known for wide range of linearity for quantified substances, why was only the range 0.5-5 ug/mL used? I think up to 100 ug/mL linearity could be achieved, thus eliminating or reducing the need for appropriate sample dilution to accommodate different phenolic acid responses.

The maximum of reliability and the most precise quantitative evaluation of unknown concentrations by interpolation (in straight-line calibration graphs), is when the measured instrument signal corresponds to a point close to the centroid of the regression line (Ideally, the calibration range should be established so that the majority of the tested sample concentrations fall towards the center of the range) (Preparation of Calibration Curves, A Guide to Best Practice, 2003); for this reason the authors preferred to use a narrower range of concentrations (0,5-5 ug/mL) and a higher number of calibration points, and to dilute the extracts, to fit the plots, in which some PAs were present in too high concentration. Moreover, the assumption of homoscedasticity (meaning “same variance”) is central to linear regression models.  Homoscedasticity describes a situation in which the error term (that is, the “noise” or random disturbance in the relationship between the independent variables and the dependent variable) is the same across all values of the independent variables.  Heteroscedasticity (the violation of homoscedasticity) is present when the size of the error term differs across values of an independent variable. In constructing a calibration plot, the condition of Homoscedasticity is highly desirable because, during the analysis, the error associated with the measurements (expressed as standard deviation of the concentration) is homogeneous for the entire interval considered. Otherwise it may happen that the error associated with the measurement of low levels of concentrations is much greater than the values of the concentration itself. As a rule, the condition of Homoscedasticity is guaranteed if the extension of the concentration of the standards is within one, two orders of magnitude maximum. In the present UHPLC method the use a concentration range between 0.5 and 5 ug/mL ensures the Homoscedasticity and the higher reliability of results.

  1. The presence of tyrosol is not confirmed in the studied material (compound is not present in Table 2) so, in my opinion, should be excluded from the study.

The authors would prefer to maintain tyrosol in the validation process, because it was investigated, even if not present in the matrix analyzed; ,moreover  it could be interesting to know its response on the UV detector and in the conditions of separation reported in this paper (eluent and gradient), its retention time…., in the event that other authors will find this compound in their matrices

  1. 11. Validation of the method of quantitative determination of free phenolic acids in BSG samples using HPLC-UV analysis.

As I wrote above, in my opinion, precisely, the validation concerns the method of determining the content of free forms of phenolic acids in BSG samples using HPLC-UV analyzes.

Ok, the text was modified (lines 231-232)

  1. Considering that both ingredients used (type of malt and hop products) and brewing process are the most important factors influencing the chemical composition of BSG and BSH by-products, I suggest to modify abbreviations used to describe BSG and BSH material in whole manuscript. Current abbreviations (barley malt vs. barley malt + wheat) fit only BSG material but not BSH (in BSH very important is hop product input). The most appropriate is probably to use abbreviations derived from type of beer: Imperial red ale beer (IRAB) vs. Belgian strong ale beer (BSAB); or other defined by authors.

Ok, thank you for the suggestion, the abbreviations were modified and the abbreviations IRA and BSA were used in the entire manuscript

  1. I suggest shortening all below values to 2 decimal places.

Ok, the table was revised

  1. As I wrote above (Table 1), the abbreviations referring only to malt (barley or wheat) are not suitable for BSH samples; they also overlook the importance of the brewing process itself; thefore I suggest changing them to a more suitable.

Ok, the table and entire text was revised and the abbreviations IRA and BSA added

  1. I suggest separating this section into 3.2 Total polyphenols and phenolic acid composition and 3.3 Antioxidant activity

The section was separated into two sections

  1. The PAs level in BSH samples is low comparing with BSG, therefore there is no basis for describing it as significant.

Ok, the discussion was modified (line 355)

  1. There is no information on the presence/content of tyrosol in studied material. If this compound is absent I suggest to exclude it from the study.

As mentioned above, the authors think that the presence of tyrosol in the validation of UHPLC-UV method could be of some utility for other authors that will analyze matrices containing this compound

  1. As I wrote above (Table 1), the abbreviations referring only to malt (barley or wheat) are not suitable for BSH samples; they also overlook the importance of the brewing process itself; thefore I suggest changing them to a more suitable.

Ok, the abbreviations were modified in the entire text and tables and figures

  1. or n=3? In section 2.4 is written that samples were analyzed in triplicate.

The replications were 6, three extraction and two analytical determination of each extract (the information was added along the manuscript (line 136)

  1. The paper 38 shows the phytochemical composition of hop cones extract but not spent hops, which probably differ in metabolite profile due to brewing process and microbial transformations. I suggest refering more to the works describing the composition of spent hops, such as https://doi.org/10.1016/j.jff.2016.01.029.

Kobus-Cisowska et al, analyzed hops, the trub is made mainly by hop residues together with other material deriving from brewing cereals, so it is possible some kind of comparison. Luzak et al, in the article suggested, analyzed the spent hops after the hop’s extraction by supercritical CO2, this material is very different from the BSH analyzed in this research paper and the authors think that to refer to this article could not applicable and not so useful.

  1. Results in Fig. 2 should be more deeply analyzed and discussed in the text.

The discussion was improved (lines 405-411)

  1. I suggest presenting IC50 data in the form of Table, it will be more readable.

In general, IC50 (Inhibition Concentration 50) values are presented with conc. units, such as mg/mL, therefore please check units. Moreover, as mentioned in 2.4.9 section, the Trolox was used as reference substance and positive control in all antioxidant assays therefore I recommend also adding IC50 values of Trolox and comparing it with studied material.

Thank you. The IC50 of Trolox was added and the results of IC50 represented in the form of graph, a discussion of graph was added along the text, the units were checked and modified.

  1. My analysis and calculations show that the IC50 values for by-products of Belgian ale were about 1.5-fold lower (better antioxidant potential) comparing with Imperial ale beer. Please try to be more precise giving conclusions and when comparing samples, because simply writing that one sample had better activity than another is definitely not enough.

Thank you for your precious comment, the discussion was improved (lines 411-427)

  1. The conclusion should be extended, did the activity results correlate with the TP content or with the PAs content? If the correlation was significant, the authors may provide the correlation coefficient.

The authors can confirm the positive correlation between TP and Antioxidant activity as well as TPA and Antioxidant activity, anyway we have only two different datapoints to correlate (FRAP values versus TP values for BSG-IRA and BSG-BSA; (FRAP value versus TPA values for BSG-IRA and BSG-BSA; DPPH values versus TP values for BSG-IRA and BSG-BSA….etc). So only a straight line can be found to interpolate two distinct points perfectly, and the only choice is between a rising and a falling straight line, which gives +1 or -1 as R, so scientifically a correlation involving only two data set and two variables is useless. (lines 425-427)

  1. As I wrote above (Table 1), the abbreviations referring only to malt (barley or wheat) are not suitable for BSH samples; they also overlook the importance of the brewing process itself; thefore I suggest changing them to a more suitable.

Thank you, The abbreviations were modified

  1. Three decimal places is probably enough precision.

Why some calibration curves were forced 0-origin (no intercept)? 

Ok, the decimal places were reduced

Three were the options to choose: “curve passes through origin”, “Ignore origin” and “Compute with origin”  The software used to calibrate the UHPLC system suggests, and automatically chooses, about the Origin (zero handling), the option “curve passes through origin” when standards with uniform amounts were used, anyway, for some plots, this choice did not allow the best fittings of the data with the curve and gave a low R, so for those plots the authors have chosen to select “ignore the origin”, to obtain the best fitting and the best R for those compounds

  1. My previous comment in first review reffered to the feed purpose (animal nutrition) of brewing by-products, in my opinion, authors should also highlight and confirm added-value of BSG and BSH in relation to feeding animals.

Ok, the authors have added a comment about the advantages of use brewing by-products in animal diet (lines 521-526)

Round 3

Reviewer 1 Report

The second version of the manuscript 1026061 is much improved, in general, the authors positively responded to most of my comments and suggestions.

General comment:

Still, I believe that the antioxidant part (especially results and discussion) can be re-written and improved by authors, to better meet Antioxidants level.

In addition, a few comments, referring mostly to the antioxidant part, are included in the attached pdf file. My current opinion is a minor revision.

Author Response

Dear Referee 1

Thank you for your contribution in the improvement of the entire manuscript

The second version of the manuscript 1026061 is much improved, in general, the authors positively responded to most of my comments and suggestions.

General comment:

Still, I believe that the antioxidant part (especially results and discussion) can be re-written and improved by authors, to better meet Antioxidants level.

The discussion about antioxidant section has been revised and improved ( Lines 401-416, lines 417-429)

In addition, a few comments, referring mostly to the antioxidant part, are included in the attached pdf file. My current opinion is a minor revision.

The comments were analyzed and manuscript revised on their bases.

  1. If I understand correctly the authors' answer to my previous comment, there were 3 extracts independently prepared and they were analyzed using HPLC in duplicate. Thus, in my opinion, n still equals 3 (independent samples or observations) because HPLC analyzes were only technical replications.

Ok, considering that the referee believes it is more proper, the analytical replications were not considered and n=3 reported (lines 135 and table 2)

  1. I suggest adding at least short explanation of the meaning 'median inhibitory conc. IC50'.

In my opinion several important details referring IC50 calculation are missing: how many different concentrations of samples and trolox were used, what was the concentration range of standard and samples, how the results were expressed (units)?

Looking at fig 3 and big difference in IC50 between trolox and samples, I wonder if IC50 results for samples were only theoretically calculated from curves or actually obtained from the measurements? When sample curve is far from actual 50% inhibition value, the calculation of sample IC50 is burdened with large error, and in this case it is probably preffered to give %inhibition of highest tested sample concentration.

The Trolox levels for the calibration plot were seven, the range of concentration between 25 uM and 800 uM.  More information were added along the text (lines 210-214)

The units of IC50 were g/l, these units were now corrected along the text (sorry there was an oversight in correcting the units of measurement in the precedent revision process- round 2).

For all three antioxidant assays (FRAP, ABTS and DPPH assays), Trolox was used as an antioxidant standard and the antioxidant evaluation and experiment procedure has been set up plotting a curve of antioxidant activity of Trolox (for each considered antioxidant assay), and the results of antioxidant capacity of analyzed samples expressed as uM of trolox/g (dm). The authors chose this procedure because interested to the antioxidant capacity of brewing by-products themselves and not of their extracts; in fact the by-product itself has a potential use as raw material for production of high valuable food/feed products, not the extracts that moreover can be obtained with different extraction procedures and solvents. Furthermore, this  expression of results (uM of trolox/g (dm)) has been used in different references on topic (Thaipong et al., 2006; Floegel et al., 2011; Zhang et al., 2013; Benincasa et al, Annals of Agricultural Sciences, 65(1), 2020).

However, following your suggestion, the authorsexpressed the results as IC50 values also (to make more clear the antioxidant potential of different by-products). Anyway, during the scientific trial, no curve of different concentrations of each sample was constructed to evaluate the IC50 of the samples, so IC50 values, of each sample and assay, have been theoretically calculated from Trolox curves.

Anyhow, the IC50 is a concentration and according to the authors it is not the most appropriate way to represent the antioxidant capacity of a solid product. Consequently the authors believe that it could more opportune to erase this expression of the antioxidant capacity from the manuscript and deeply comment the results expressed as uM of trolox/g (dm).
